# Effect of Different Surface Treatments on the Shear Bond Strength of Metal Orthodontic Brackets Bonded to CAD/CAM Provisional Crowns

**DOI:** 10.3390/dj11020038

**Published:** 2023-02-02

**Authors:** Dany Haber, Elie Khoury, Joseph Ghoubril, Nunzio Cirulli

**Affiliations:** 1Department of Orthodontics, Faculty of Dental Medicine, Saint Joseph University, Beirut 10999, Lebanon; 2Department of Interdisciplinary Medicine, University of Bari Aldo Moro, 70100 Bari, Italy

**Keywords:** CAD/CAM provisional crowns, surface treatments, shear bond strength

## Abstract

Background: The aim of this study was to find the best surface treatment for CAD/CAM provisional crowns allowing the optimal bond strength of metal brackets. Methods: The sample consists of 30 lower bicuspids and 180 provisional crowns. The provisional crowns were randomly divided into six different groups. Orthophosphoric acid etching (37%) was applied to 30 lower bicuspids. The provisional crowns had undergone different surface treatments. Group 1: No treatment (Control Group). Group 2: Diamond bur. Group 3: Sandblasting. Group 4: Plastic Conditioner. Group 5: Diamond bur and Plastic Conditioner. Group 6: Sandblasting and Plastic Conditioner. The brackets in all groups were identically placed using Transbond XT^®^ Primer and Transbond XT^®^ Paste. Then, the entire sample underwent an artificial aging procedure, and a measurement of the bond strength was conducted. After debonding, the surface of the crowns was examined to determine the quantity of the adhesive remnant. Results: Bonding to natural crowns recorded the highest average, followed by the averages of groups 5 and 6. However, group 1 recorded the lowest average. Groups 2 and 4 had very close averages, as well as groups 5 and 6. A statistically significant difference between the averages of all groups was recorded (*p* < 0.001) except for groups 2 and 4 (*p* = 0.965) on the one hand, and groups 5 and 6 (*p* = 0.941) on the other hand. Discussion: The bonding of brackets on provisional crowns is considered a delicate clinical procedure. In fact, unlike natural crowns, the orthophosphoric acid usually used does not have any effect on the surface of provisional crowns. Conclusions: Using a diamond bur combined with the plastic conditioner and sandblasting combined with that same product resulted in a bond strength close to natural crown.

## 1. Introduction

Nowadays, orthodontics is attracting young and adult patients who both demand a perfect smile [1]. In some cases, a provisional crown is deemed necessary to maintain the functional, esthetic, and therapeutic role of teeth on the dental arch [2,3]. In fact, in some orthodontic treatments, the brackets must be placed on provisional crowns. Yet, as is the case with natural crowns, the brackets must have a bond strength which is enough to resist orthodontic and functional forces.

There are many types of provisional crowns, and each type has distinctive characteristics and properties [2,3]. Even though none of them is qualified as “ideal material”, provisional crowns made in the laboratory have proven to be advantageous.

Today, CAD/CAM dentistry, which can produce provisional crowns, is one of the most successful of laboratory technologies [4]. However, bonding of orthodontic brackets to provisional crowns is a challenge. Thus, choosing the appropriate surface treatment is of prime importance.

The null hypothesis is that applying plastic conditioner, which is an adhesion promoter, improves the bond strength on this type of provisional crowns.

In order to reduce the rate of debonded brackets, and therefore the duration of orthodontic treatment, this study aims to find the best surface treatment for provisional crowns made in the laboratory using the CAD/CAM system allowing the optimal bond strength of brackets.

## 2. Materials and Methods

### 2.1. Sample

For periodontal or orthodontic reasons, 30 lower bicuspids were freshly extracted and selected, having an intact buccal side of their crown.

One of the 30 teeth was scanned using the Ceramill map 400 to make 180 provisional teeth by means of the milling machine, the Ceramill Motion 2 run by the CAD/CAM system (Figure 1).

All teeth were inserted in acryl blocs which were cooled down in a special mold. The teeth were inserted using a surveyor device so that the teeth axis is homogenous for the entire sample. Lower bicuspids metal brackets (Mini Twin (Ormco Corporation 200 S Kraemer Blvd, Brea, CA 92821 USA)) were used [5]. One operator prepared the sample.

### 2.2. Measurement Methods

#### 2.2.1. Natural Teeth (NT)

Orthophosphoric acid etching (37%) (Biodinamica R. Ronat Valter Sodré, 4350 Parque Industrial, Ibiporã–PR, 86200-000 Brasil) was applied for 15 s to the buccal side of the teeth. The acid was eliminated with water and the teeth were dried properly. A thin layer of Transbond XT^®^ Primer (3M Unitek, Irwindale, CA, USA) adhesive was applied. The brackets were placed in a standard position following the axis of each tooth, at a 4 mm height measured from the edge with the Transbond XT^®^ Paste adhesive (3M, Unitek) using a gauge. A constant force of 5 Newtons was exerted on every bracket for 5 s using a tensiometer (Bongshin, Osan, Republic of Korea, Model BS-201 Series) to obtain a uniform adhesive layer thickness. The excess adhesive around the bracket base was removed and the photopolymerization was applied for 10 s on the mesial side and 10 s on the distal side [6].

#### 2.2.2. Provisional Teeth

The 180 provisional teeth were randomly divided into 6 groups (*n* = 30). Different surface treatment was used for each group.

Group 1 (Control group)

No surface treatment was applied.

Group 2 (Diamond bur)

The buccal surface was roughened using a diamond cylindrical bur (PacDent, 670 Endeavor Cir, Brea, CA 92821, United States) mounted on a handpiece with a rotation speed of 2000 rpm. This bur was placed parallel to the surface of the tooth and brushing movements were made for 10 s. The tensiometer machine was also used to standardize the force applied to the tooth (maximum force of 1 Newton) during milling. An observer was watching the screen of the tensiometer machine to avoid exceeding the maximum value when handling.

Group 3 (Sandblaster)

This group was treated with a sandblaster to create microporosities on the buccal surface of the provisional restorations by applying particles of 50 μm aluminum oxide.

The sandblasting process was carried out perpendicularly to the tooth surface at 1 cm for at least 5 s until the surface of the restorations became smooth. The excess of aluminum oxide was removed with an air jet [7].

Group 4 (Plastic Conditioner)

A uniform thin layer of Plastic Conditioner (Reliance Orthodontic Products 1540 W Thorndale Ave, Itasca, IL 60143, USA) was placed on the buccal surface of the provisional restorations. After 60 s, the bracket bonding process was carried out [8].

Group 5 (Diamond bur + Plastic Conditioner)

The same protocol performed in groups 2 and 4 was applied to this group; milling of the buccal surface of provisional restorations then applying the product Plastic Conditioner to them.

Group 6 (Sandblaster + Plastic Conditioner)

The same protocol performed in groups 3 and 4 was applied to this group; sandblasting of the buccal surface of provisional restorations then applying the product Plastic Conditioner to them.

After performing the various surface treatments, bracket bonding was carried out identically in all the groups using the Transbond XT^®^ Primer and the Transbond XT^®^ Paste adhesive in the same way usually used on natural teeth.

In order to standardize the placement and bonding of brackets, a transparent thermoplastic mould was specifically manufactured. The latter was designed based on the natural tooth, which served to fabricate the provisional teeth, after bonding the bracket (Figure 2).

The entire sample underwent an artificial aging procedure (“SD Mechatronik THERMOCYCLER” machine). This procedure consists of an exposure to 2200 cycles which is equivalent to the average duration of an orthodontic treatment (18–20 months) [9]. The cycle includes 30 s of immersion in a bath of cold water (5 °C) followed by an immersion in a hot water bath (55 °C) separated by an interval of 10 s in the open air.

The measurement of the bonding force was carried out for the whole sample using the Universal Testing Machine YLE (GmbH Waldstrasse 1/1a, YLE GmbH Waldstrasse 1, D-64732 Bad Koenig, Germany) connected to a computer (Figure 3).

The shear bond strength in megapascals (MPa) was calculated by dividing the fracture load in Newton by the surface area of the bracket in square millimeter (8 mm^2^).

After debonding, the surface of the teeth was examined by two observers independently using an optical microscope (Karl Kaps Optik-Feinmechanik-Gerätebau GmbH & Co. KG Europastraße, 35614 Aßlar, Germany) under different magnifications in order to determine the amount of adhesive remnant based on the Artun J and Bergland S Index (1984) [10].

−0 = no adhesive left on the tooth (Figure 4).−1 = less than half of the adhesive left on the tooth (Figure 5).−2 = more than half of the adhesive left on the tooth (Figure 6).−3 = all adhesives left on the tooth with a distinct impression of the bracket mesh (Figure 7). It should be noted that scores of 0 and 1 imply a fracture of the bond at the level of the tooth/adhesive interface (adhesive failure). Scores 2 and 3 involve a fracture at the adhesive/bracket interface (cohesive failure).

**Figure 4 dentistry-11-00038-f004:**
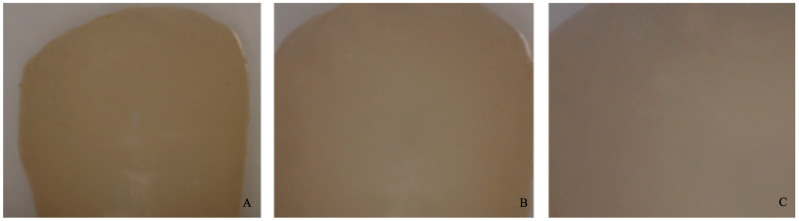
Score 0. (**A**) Magnification: 10 × 1; (**B**) Magnification: 10 × 1.6; (**C**) Magnification: 10 × 2.5.

**Figure 5 dentistry-11-00038-f005:**
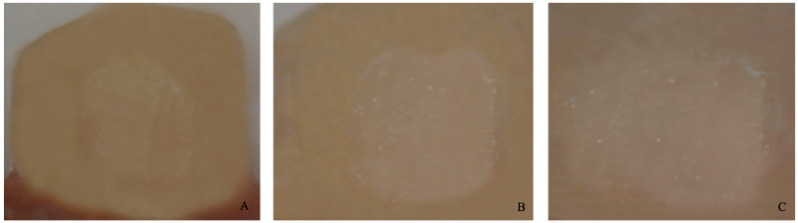
Score 1. (**A**) Magnification: 10 × 1; (**B**) Magnification: 10 × 1.6; (**C**) Magnification: 10 × 2.5.

**Figure 6 dentistry-11-00038-f006:**
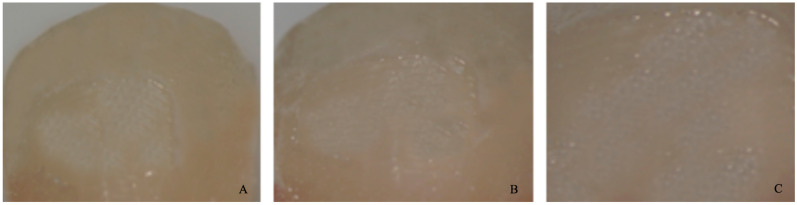
Score 2. (**A**) Magnification: 10 × 1; (**B**) Magnification: 10 × 1.6; (**C**) Magnification: 10 × 2.5.

**Figure 7 dentistry-11-00038-f007:**
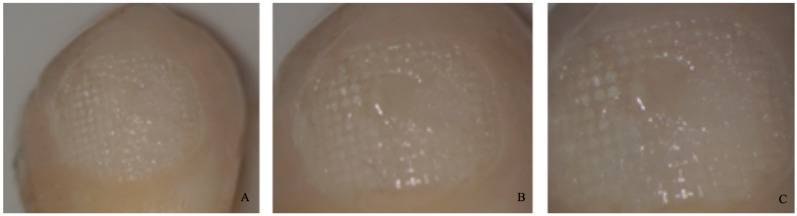
Score 3. (**A**) Magnification: 10 × 1; (**B**) Magnification: 10 × 1.6; (**C**) Magnification: 10 × 2.5.

All samples are standardized: type of bracket, amount of bonding adhesive, and debonding method.

### 2.3. Statistical Analysis

The statistical software SPSS (Statistical Package for Social Science) for Microsoft Windows version 23.0 was used for the statistical analysis of data. The level of significance was set at 0.05 (*p* ≤ 0.05).

The normality of distribution of bond strength values was tested by the Shapiro–Wilk test and the homogeneity of the variance was tested using the Levene test.

An analysis of variance (ANOVA) modified according to Welch was used to determine any significant difference in the bond strength between the seven groups. A post hoc Tukey test was used to identify which means were significantly different from each other.

Two different observers evaluated the ARI index. The Chi-square test and the Cramer’s V test were used to determine the statistical significance of the ARI scores.

For each group, the Pearson correlation coefficient was established between the bond strength and the ARI index.

## 3. Results

Bond strength:

The highest average was observed in natural crowns (6.58 ± 1.09 MPa) followed by the averages of groups 5 and 6 (5.15 ± 0.68 MPa and 5.35 ± 0.66 MPa respectively). The lowest average was observed in group 1 (0.48 ± 0.73 MPa).

The averages of groups 2 and 4 (2.89 ± 0.56 MPa and 2.71 ± 0.44 MPa respectively) yielded close results. The same goes for groups 5 and 6 (5.15 ± 0.68 MPa and 5.35 ± 0.66 MPa respectively) (Table 1).

There is a normality in the distribution of bond strength (*p*-value < 0.001), hence the possibility to use parametric statistical tests. There is also a lack of homogeneity in the variance between the groups (*p* < 0.001).

A significant statistical difference is present between the averages of the different groups (*p* < 0.001). When comparing the averages of the different groups, it appears that there is a significant difference between the averages of all the groups (*p* < 0.001) except for groups 2 and 4 (*p* = 0.965) on one hand, and groups 5 and 6 (*p* = 0.941) on the other.

The results of Table 2 showed that groups Natural croens, 3, 5, and 6 yielded scores that are significantly higher than those of groups 1, 2, and 4. Group 1 mainly yielded a score equal to 0, while groups 2 and 4 yielded a score equal to 1. However, groups Natural crowns, 3, 5, and 6 are predominated by score 2. The ARI significantly differs when comparing the groups (*p* < 0.001).

Quantity of residual adhesive (ARI):

Table 2 shows that groups Natural crowns, 3, 5, and 6 are similar as well as groups 1, 2, and 4. Nevertheless, the ARI score is statistically different when comparing groups Natural crowns, 3, 5, and 6 on one hand, and groups 1, 2, and 4 on the other.

Correlation between bond strength and ARI:

There is a strong proportional correlation between the means of bond strength and the ARI in group 1 (*r* = 0.689). This correlation proved to be very strong and proportional in groups Natural crowns, 2, 3, 4, 5 and 6 (*r* = 0.840).

## 4. Discussion

Over the next decade, as dentists become more familiar with modern technologies, such as CAD/CAM systems, we can expect an increase in the use of this technique in dentistry [11]. The CAD/CAM system can produce crowns from blocks of a material called “Polymethyl methacrylate” (PMMA). This digital procedure reduces several obstacles that are present in the process of making direct provisional crowns (better mechanical and esthetic properties). As far as we know, few studies have been carried out on bonding brackets to provisional crowns made in the laboratory using the CAD/CAM system. They were mainly carried out on light-cured resins [12,13], acrylic resins [6,14] and prefabricated resins [15,16]. For all these reasons, it was therefore interesting to conduct a study on the new emerging technologies.

However, we worked on the lower bicuspid teeth since the fracture rate of brackets is the most important in the mandibular lateral sectors [17,18,19].

During an orthodontic treatment, the bond strength of the brackets could depend on several factors including the used material and the treatment surface. The bonding of orthodontic brackets can be performed on different surfaces: whether on enamel or dentin which are natural substrates, or on restorations such as composite, and even on provisional crowns, which are not natural substrates. For this reason, bonding procedures differ from a surface to another. However, the surfaces of provisional crowns made in the laboratory will be subjected to intensive polymerization and heat treatment to increase their mechanical properties. So, this polymerization will limit the possibilities of bonding since the surface of the provisional crowns will lose the function of creating a strong bond with the adhesive. In order to solve this problem, the surface of the crowns must undergo a surface treatment to increase the bond strength.

In fact, unlike natural teeth, the orthophosphoric acid usually used does not have any effect on the surface of provisional restorations. It only allows eliminating the impurities left behind by the sandblasting process [14]. For this reason, several surface treatment techniques were proposed to enhance the bond strength between the orthodontic brackets and the surface of the provisional restorations. The literature showed that dental milling and sandblasting both increased the bond strength between the brackets and the surface of the provisional restorations [20,21]. Nevertheless, several studies comparing these two techniques showed that the surface of the provisional restorations becomes rougher after a sandblasting procedure, thus ensuring better bond strength [14,20,22]. According to Blakey R et al., the bond strength improved but remained inferior to the optimal force on natural crowns [20]. This is in line with the results of our study.

In order to enhance the bond strength on the provisional crowns, the application of a product, the Plastic Conditioner, was proposed. This product is made of Methyl Methacrylate (a monomer) and Isobutyl Methacrylate. It is an adhesion promoter that chemically links the surface of the provisional restoration to the bonding product.

Several studies were performed to examine the effectiveness of this product. The study conducted by Egan FR et al. showed that using this product on natural teeth did not allow improving the adhesion force of the rebonded brackets [23]. The study conducted by Masioli et al. on provisional restorations showed that this product formed a layer preventing contact between the bracket and the provisional restoration, thus compromised the bond strength [14]. A study conducted by Tse in 2012 to evaluate the bond strength on restorative composite proved that applying Plastic Conditioner alone is not enough [8]. When coupling it with milling, the bond strength improved. This study yielded similar results to ours, as the use of the product alone is not beneficial, but when coupling it with milling or sandblasting, the bond strength improved and was close to the bond on natural teeth. This can be explained by the fact that the Plastic Conditioner managed to penetrate the roughened surface resulting from the milling and sandblasting procedures.

All the possible surface treatments for the provisional crowns have been evaluated in this study except the application of 9.6% of hydrofluoric acid. It has been proven that this is ineffective [9].

There are many bonding materials in orthodontics. The most used are acrylic resins, which belong to the chemical polymerized bonding material, and the Transbond XT^®^ which belongs to the light cured bonding material.

Dias et al. and Almeida et al. found in their studies that acrylic resins are better than Transbond XT^®^ for bonding brackets on acrylic surfaces [6,22]. These studies agree with Zachrisson et al., who recommend using acrylic resins for bonding on acrylic surfaces since their adhesion force is higher than that of other adhesives [24].

Using acrylic resins is theoretically recommended [25]. However, practically, acrylic resins have some disadvantages, such as their handling, which is much harder than that of Transbond XT^®^. Besides, after bonding the brackets to the teeth, the practitioner should wait 5 min before inserting any orthodontic wire. For all these reasons, we decided in our study to find the best surface treatment that would allow us to reach the best bonding strength of brackets, which is by using the Transbond XT^®^ adhesive.

During an orthodontic treatment, which can extend from approximately 18 to 24 months, the bonding strength of brackets decreases with time [25]. For this reason, it would be interesting to simulate the aging procedure “in vitro” in a thermocycling machine. The International Organization for Standardization (ISO) indicates that it would be appropriate to perform 500 cycles in water between 5 °C and 55 °C separated by an interval of 5 s for a suitable aging simulation (ISO 11405) [26]. However, this number of cycles is probably insufficient to carry out a “real” simulation [27]. Recent studies in orthodontics have increased the number of cycles up to 1500 cycles between 10° and 50 °C [28] and 6000 cycles between 5 °C and 55 °C [29].

In a study by Reicheneder et al. in 2007, a simulation of 9–10 months was performed for a duration of three days where brackets were exposed to 1100 alternating cycles of hot water baths (55 °C) and cold water (5 °C) [9]. To perform an aging simulation of 18–20 months, the same procedure can be followed but with 2200 cycles and it lasts six days. Therefore, we chose in our study a total of 2200 cycles, corresponding to an average duration of an orthodontic treatment. In addition, the choice of the composition of each cycle is based on the ISO 11405 recommendation: 30 s in each of the baths of 5 °C and 55 °C separated by an interval of 10 s in the open air.

In order to choose the right orthodontic adhesive, the evaluation of the amount of residual adhesive after debonding is essential and it can be assessed with ARI scoring system [30]. To be able to compare our results with other studies [6,20,22,25], we used the original method (Artun J. and Bergland S. Index, 1984) instead of the modified one (Bishara amd Trulove, 1990). A direct comparison between these two ARI scoring methods could not be made since the number of scores is not similar [31]. 

The predominant ARI score for groups 2, 3, and 4 was 1, in opposition to groups 5 and 6 where the predominant ARI was 2. This can be explained by the fact that the Plastic Conditioner penetrates the microporosities produced by the diamond bur or sandblaster to yield a better chemical interaction and a higher bond strength [14]. The only disadvantage of this type of cohesion is the high time consumption required to eliminate the entire quantity of the residual adhesive [30].

One of the limitations of this study is that the evaluation of the ARI was performed visually under microscope (KAPS, Germany) under different magnifications. This will make it difficult to obtain reliable measurements.

The other limitation is the oral environmental factors that can affect the bond strength. It has been proven that the mean bond strength values in vivo are 40% less than in vitro [32,33]. For this reason, further work is required to explore the differences between the bond strengths of different surface treatments in the oral environment.

## 5. Conclusions

The two groups in which we used a diamond bur and the Plastic Conditioner as well as sandblasting and the Plastic Conditioner exhibited bond strengths which were very close to those of natural teeth. Since there was no statistically significant difference between these two groups, it would be easier and more specific to perform sandblasting of the provisional crown.

Finally, this study can help orthodontists to affectively choose the more ideal surface treatment, when possible, to avoid the debonding of the brackets and reduce the orthodontic treatment period.

## Figures and Tables

**Figure 1 dentistry-11-00038-f001:**
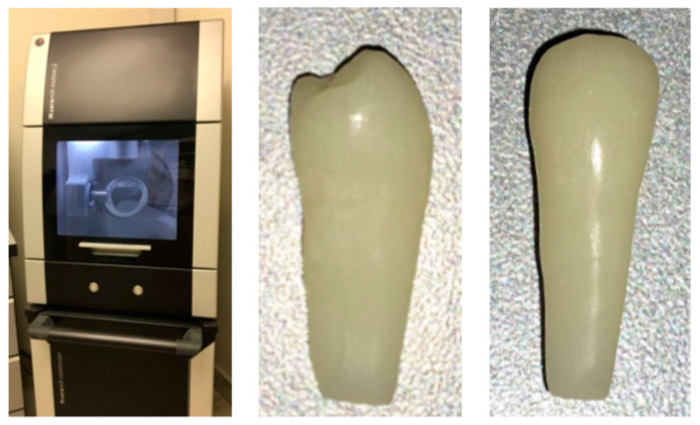
The Ceramill Motion 2 machine and one of the provisional teeth.

**Figure 2 dentistry-11-00038-f002:**
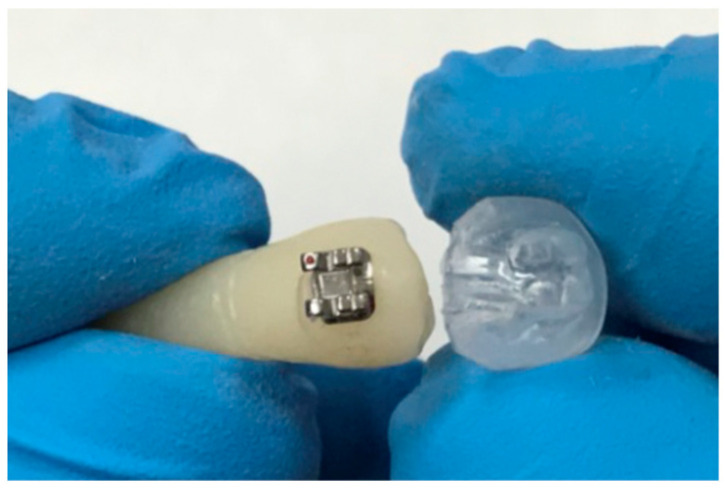
Transparent thermoplastic mould specially designed to standardize the placement and bonding of brackets.

**Figure 3 dentistry-11-00038-f003:**
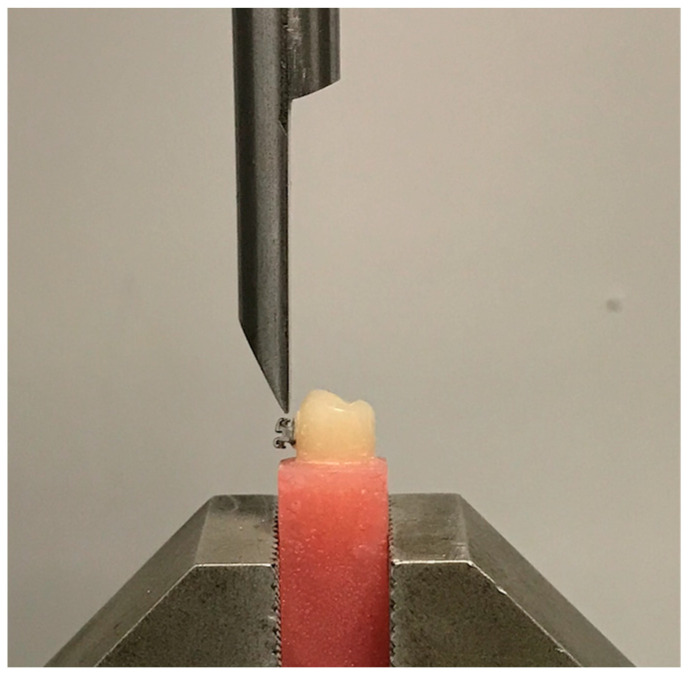
The Universal Testing Machine YLE (GmbH Waldstraße 1/1a, YLE GmbH Waldstrasse 1, D-64732 Bad Koenig, Germany).

**Table 1 dentistry-11-00038-t001:** Means of bond strength for the different groups.

Group	N	Average	Standard	Confidence
		(in MPa)	Deviation	Interval at 95%
NT	30	6.58	1.09	6.98–6.17
1	30	0.48	0.73	0.75–0.21
2	30	2.89	0.56	3.10–2.68
3	30	4.21	0.82	4.52–3.70
4	30	2.71	0.44	2.88–2.55
5	30	5.15	0.68	5.40–4.89
6	30	5.35	0.66	5.59–5.10

**Table 2 dentistry-11-00038-t002:** Scores of the quantity of residual adhesive (ARI) for each group.

Group	N	Scores of the Quantity of Residual Adhesive (ARI), %			
		0	1	2	3
NT	30			70	30
1	30	90	10		
2	30	10	70	20	
3	30		36.7	60	3.3
4	30	23.3	56.7	20	
5	30		30	63.3	6.7
6	30		20	56.7	23.3

## Data Availability

All data generated or analyzed during this study are included in this published article. All experimental data to support the findings of this study are available by contacting the authors.

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
