# Peer review of "Effect of Different Surface Treatments on the Shear Bond Strength of Metal Orthodontic Brackets Bonded to CAD/CAM Provisional Crowns"

_dentistry, 2023, doi:10.3390/dj11020038_

Round 1
Reviewer 1 Report
Orthodontic treatment is an important clinical technique for which demand is gradually increasing.
This study sought conditions to create optimal surface conditions for brackets using provisional crowns.
There are many studies on the bonding strength of general restorative resins.
However, research on the bonding strength of brackets using provisional crowns is limited compared to that.
In order to improve the satisfaction of orthodontic treatment patients and dentists, it is considered that the bonding strength of brackets made of various materials is important.
(1) Title: Please add the term “bracket”, which is the main dental material used in the study.
(2) Introduction:
- Please add the meaning of “provisional crown” and “plastic conditioner”.
- Although studies using provisional crowns are limited, related previous studies should be presented.
- In the Materials and methods there is information about “Ceramill Motion 2” of CAD-CAM. Please add a background explanation.
(3) Materials and methods:
- There are many figures in the manuscript. I think you can delete Figure2.
- Rather than Group 1, 2, 3, 4, 5, 6, it is better to increase understanding with intuitive terms. (diamond bur, sandblaster, plastic conditioner, etc.)
- The experimental method of Artun J and Bergland S Index (1984) (9) is an old experimental method. Are there any alternative test methods? Please write the limitations of the research method in the discussion.
- Figure 4 photo is not visible in detail. A high-resolution, easy-to-understand picture would be useful.
- Please add information that all samples are standardized, such as type of bracket, amount of bonding adhesive, and debonding method.
It is easy to see at a glance for merging Figures 5 with 8 suggest
Lines 162-176: please delete the Author Guidelines.
(4) Results
- Table 1. Please write a detailed explanation of Groups 1-6 in the footnote.
please express as ±SD in the table.
(5) Discussion
A study on the surface treatment method and resin combination of provisional crowns has been reported. Please write a discussion by referring to the manuscript below.
Effect of surface treatments and fash‑free adhesive on the shear bond strength of ceramic orthodontic brackets to CAD/CAM provisional materials. Clinical Oral Investigations (2022) 26:481–492.
Shear Bond Strength of Orthodontic Brackets Bonded to Provisional Crown Materials Utilizing Two Different Adhesives. Angle Orthod (2009) 79 (4): 784–789.
The limitations of the study are missing. Please add limitations and future research points.
(6) References
Please modify all of the following according to the submission policy.
Pabari, S.; Moles, D.R.; Cunningham, S. Assessment of motivation and psychological characteristics of adult orthodontic patients. Am. J. Orthod. Dentofacial. Orthop. 2011, 140, 201-205.
Author Response
Thank you for having found that the work is very interesting.
An extensive English revision will be done to the manuscript.
- Title:
The title was modified to let the reader understand that this study is dealing with the surface treatment of CAD/CAM provisional crowns and its effect on the shear bond strength of the brackets.
- Abstract:
We tried to unify the use of terms in the abstract and in the entire document.
- Introduction:
- We did a logic links between the ideas.
- We talked about the CAD-CAM provisional crowns (material they are made of…) in the discussion.
- Some citations were added for statements that we made.
- Materials and methods:
The paragraph explaining why we did not choose the hydrofluoric acid was moved to the discussion part.
- Results:
- We mentioned that the values presented in the text are in Megapascals and we explained how they were calculated.
- All statistical tests were removed.
- Discussion:
We talked about the material from which the provisional crowns were made and we explained what is the Plastic Conditioner, what it contains and how it works.
- Conclusion:
The discussion part was removed, and the conclusion is now focusing on the study's general objective.
Reviewer 2 Report
Dear Authors
With great interest and emotion, I have read your article "Evaluation of the bond strength of provisional crowns made in the laboratory (CAD/CAM)." In general, the manuscript provides important results, but it is not well written. A professional language review would help to order your ideas and make the manuscript more understandable. In addition, I have a few concerns and suggestions that must be attended to. Next, I will give you the most important ones.
Title
The title is not the most appropriate; at first glance, it makes the reader believe that the study deals with the bond strength of the provisional crowns to teeth. As if what was to be tested was the cementation of the provisional crown to the tooth.
The title should mention "Surface treatment" and "braquets," at least.
If the title reads "made in the laboratory (CAD/CAM)," it leaves many doubts; is that the name of your laboratory?
It is better to include the name of the material you tested.
Abstract.
It is not descriptive enough; it is confusing and needs to be restructured.
There must be consistency in the terms. The clearest example is (and this applies to the entire document, not just the abstract) that you call CAD/CAM fabricated crowns differently.
"provisional crowns"
"provisional restorations"
"Provisional teeth" this last is also the most confusing. What is a provisional tooth? At first, I thought you were referring to a temporary tooth.
Introduction
The first paragraph (lines 30-32) is unrelated to the second one (lines 32-34). Being an adult does not mean that you necessarily require provisional restorations. It is not even clear what you mean by "in such clinical cases" (line 34).
Writing on line 40: .. "CAD/CAM dentistry," or on line 47:" made in the laboratory (CAD/CAM)" is not precise, as well as in the title. CAD/CAM means: CAD (Computer-aided design) and CAM (computer-aided design manufacturing). Instead of telling the reader the technology that was used to manufacture the "provisional teeth" (which is not a correct term either because there are no "provisional teeth" and you could rather call them "artificial teeth," to be the opposite of what you call "natural teeth"), you should make it clear what material they are made of, even the conditions of their manufacture would be necessary.
Some citations are needed for statements that you make. For example Lines 37-39 and line 40.
On line 53, you call it CAD/CAM "system," then on line 212, you call it CAD/CAM "Software" unify and always call it the same.
Materials and methods
Please delete the paragraph on lines 105-107. This section is not intended to explain why you chose or did not choose to evaluate a product or protocol. In order to help you to remember what this section (materials and methods) should include, please read lines 162-176, and then delete them from the manuscript.
And please also delete lines 207-209 as well. It is crucial that before pressing the "submit" button, you read the document carefully; in this way, you can be sure of what the reviewer will receive.
Results
It is necessary to mention that the values presented in the text are megapascals, and it should be indicated how they were calculated. Which is the size of the brackets?
In the results, it is unnecessary to repeat which statistical tests were performed; you already mentioned that in detail in the Statistical Analysis section.
Discussion
A deeper discussion is needed.
It is necessary to discuss the material from which the "artificial" teeth were made and the reasons for good or bad adherence to it with the different protocols, explaining why some can have more effect than others.
It is also essential to write about the "Plastic conditioner"; what is that? What does that contain? What is the working mechanism? In no section of the manuscript do you mention anything about this product, and, like the material you used to make the teeth, they are the focus of the investigation.
Conclusion
All the text in the conclusion section is part of the discussion. Limit yourself to writing in the conclusion a text aligned with the study's general objective.
And a citation is needed in the statement you make in lines 267-269
I repeat it, the work is very interesting, but the quality of the written document needs to be significantly improved.
Author Response
Thank you for your valuable remarks.
- Title:
We added the term bracket in the title.
- Introduction:
- We explained more in the entire document about the meaning of “provisional crown’’ and ‘’Plastic Conditioner’’.
- We mentioned previous studies about the provisional crowns in the discussion.
- Materials and methods:
- We did not talk about the ‘’Ceramill Motion 2’’ of CAD/CAM because it’s a milling machine and it does not affect the materials used to fabricate the provisional crowns. However, we mentioned that it’s a milling machine in the materials and methods.
- We deleted Figure 2.
- Intuitive terms like diamond bur, sandblaster and Plastic Conditioner were added to the groups to be more clarified.
- The majority of the studied used Artun J. and Bergland S. index to evaluate the quantity of the residual adhesive and to be able to compare our results with other studies, we used the original method (Artun J. and Bergland S. Index, 1984) instead of the modified one (Bishara amd Trulove, 1990): we talked about this idea in the discussion.
- We added the information that all samples are standardized such as type of brackets, amount of bonding adhesive, and debonding methods.
- Figures 4 to 7 were merged to make it easy to see.
- Results:
In tables 1 and 2, intuitive terms like diamond bur, sandblaster and Plastic Conditioner were added to the groups to be more clarified.
- Discussion:
- The discussion was modified and reorganized according to a recent study: “Effect of surface treatment and fash-free adhesive on the shear bond strength of ceramic orthodontic brackets to CAD/CAM provisional materials. Clinical Oral Investigations (2022) 26: 481-492.”
- We added 2 limitations in this study:
1- The visual evaluation of the ARI index.
2- The oral environmental factors.
- References:
All the references were modified according to the submission policy.
Round 2
Reviewer 2 Report
You have done a good job, now the manuscript is of quality. Thank you very much.